# Fine root and soil carbon stocks are positively related in grasslands but not in forests

Avni Malhotra [1,2,3] ✉, Jessica A. M. Moore[4], Samantha Weintraub-Leff [5], Katerina Georgiou[6,7], Asmeret Asefaw Berhe [8], Sharon A. Billings[9], Marie-Anne de Graaff[10], Jennifer M. Fraterrigo [11], A. Stuart Grandy[12], Emily Kyker-Snowman[13], Mingzhen Lu[14], Courtney Meier [5], Derek Pierson [15], Shersingh Joseph Tumber-Dávila [1,16], Kate Lajtha [17], William R. Wieder [18] & Robert B. Jackson [1,19]

Increasing fine root carbon (FRC) inputs into soils has been proposed as a solution to increasing soil organic carbon (SOC). However, FRC inputs can also enhance SOC loss through priming. Here, we tested the broad-scale relationships between SOC and FRC at 43 sites across the US National Ecological Observatory Network. We found that SOC and FRC stocks were positively related with an across-ecosystem slope of 7 ± 3 kg SOC m$^{-2}$ per kg FRC m$^{-2}$, but this relationship was driven by grasslands. Grasslands had double the across-ecosystem slope while forest FRC and SOC were unrelated. Furthermore, deep grassland soils primarily showed net SOC accrual relative to FRC input. Conversely, forests had high variability in whether FRC inputs were related to net SOC priming or accrual. We conclude that while FRC increases could lead to increased SOC in grasslands, especially at depth, the FRC-SOC relationship remains difficult to characterize in forests.

Increasing and deepening root inputs into soils is proposed as a mechanism to increase soil organic carbon (SOC), but it remains unclear to what extent and under which environmental conditions this will be an effective strategy[1,2]. Fine roots (typically defined as roots with <2 mm diameter) are a key input to SOC and contribute disproportionately to SOC formation[3–5]. However, experimental evidence suggests that fine roots can either stabilize or destabilize SOC[6–9]. On one hand, labile compounds released by rhizodeposits or root litter may increase microbial biomass and more critically, microbial necromass, thus increasing soil organic matter (SOM) if this necromass is stabilized by minerals[4,10,11]. On the other hand, the release of labile compounds from roots may cause a net loss of SOC (priming) by the breakdown of chemical associations between organic compounds and reactive soil minerals[12], and by stimulating microbial respiration of detrital carbon (C)[13–15].

Whether fine root C inputs drive SOC accrual or priming is expected to vary by ecosystem and vegetation type, soil moisture, SOC stock and its distribution between particulate and mineral-associated pools, the amount and reactivity of soil minerals, and macro and micro soil nutrients[6–8]. Since these factors vary throughout the organic and mineral layers of soils, we also expect soil depth and horizons to be important predictors of the relationship between fine root C and SOC[16,17]. Ecosystem types with different dominant vegetation could also vary in their accrual and priming behaviors due to differences in belowground allocation, rooting depth, and other root

traits[18–20]. One emerging hypothesis is that SOC accrual is highest in soils with high reactivity minerals and in high moisture conditions, where plant productivity, SOM transport through the profile, and SOM stabilization to mineral surfaces are also high[13,21,22]. However, this hypothesis remains untested with regard to root-derived organic matter inputs. Quantifying the long-term stabilization of fine root C into SOC requires repeat measurements over multiple decades. Given the lack of such datasets[23–27], natural gradients spanning variation in soil and ecosystem types, and containing soil and root measurements across depths, provide one means of testing the long-term, steady-state relationship between fine root C and SOC.

Here, we used a natural gradient with varying fine root biomass carbon stocks (hereafter, FRC) to explore the relationship between FRC and SOC stocks. We also tested how the FRC-SOC relationship varies by ecosystem type, soil depth, and soil horizon (organic or mineral), and how the relationship is influenced by climate, mineralogy, and soil nutrients. We expected grassland ecosystem types to have stronger FRC-SOC relationships than forests because of the high below:aboveground biomass ratio in grasslands[19,20]. We also expected that mineral horizons would have a stronger relationship between FRC and SOC than organic horizons because the latter are likely more influenced by aboveground litter inputs[28]. Additionally, in mineral horizons where SOC stabilization can proceed via organo-mineral interactions, FRC is likely linked to net SOC to a greater extent than in organic horizons, where SOM may be more prone to decomposition[5].

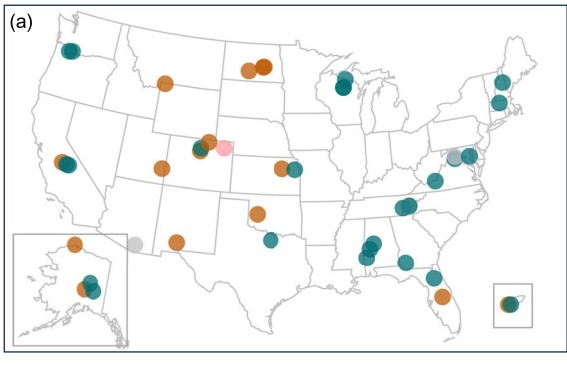
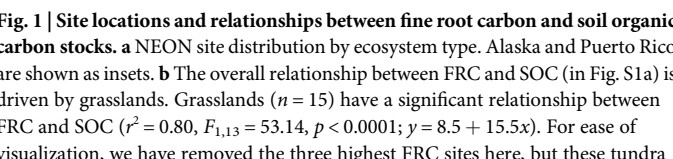
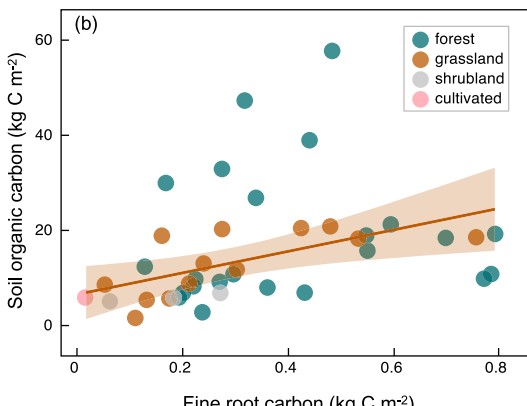

**Fig. 1 | Site locations and relationships between fine root carbon and soil organic carbon stocks. a** NEON site distribution by ecosystem type. Alaska and Puerto Rico are shown as insets. **b** The overall relationship between FRC and SOC (in Fig. S1a) is driven by grasslands. Grasslands ($n = 15$) have a significant relationship between FRC and SOC ($r^2 = 0.80$, $F_{1,13} = 53.14$, $p < 0.0001$; $y = 8.5 + 15.5x$). For ease of visualization, we have removed the three highest FRC sites here, but these tundra sites are included in Fig. S1a and Table S2. Removing these outliers decreases the $r^2$ to 0.46 but increases the slope to 22.6 kg SOC FRC$^{-1}$ m$^{-2}$. Forest FRC and SOC are unrelated ($n = 25$). Shrubland ($n = 3$) and cultivated ($n = 1$) ecosystem types do not have adequate sample sizes to analyze FRC-SOC relationships. Table S2 also provides information on depth of sampling, typically 2 m.

Furthermore, by comparing our observed FRC-SOC relationships with a theoretical one-to-one relationship between FRC input and SOC stock, we inferred net SOC accrual versus priming across the gradient. Specifically, we assumed that sites with observed SOC above the 1:1 line of standardized SOC and FRC data indicate the potential for net SOC accrual (hereafter, SOC accrual), whereas sites with SOC below the 1:1 FRC-SOC line indicate the potential for net priming (hereafter, priming). We hypothesized that SOC accrual would be highest in ecosystems with high moisture and clay content, where plant production and mineral stabilization of fine root litter would be optimized[13,21]. Conversely, priming would be more likely in ecosystems with lower moisture and clay content, where SOC would have a lower probability of interacting with soil minerals. We also hypothesized that SOC accrual would be greater at depth due to higher concentrations of reactive minerals and/or metals and lower microbial abundance/activity than in surface soils[29,30].

We used the continental gradient provided by the National Ecological Observatory Network (NEON; Fig. 1a), where coupled FRC and SOC measurements to 2-m depth have been conducted at 43 sites across the USA (see Table S1 for data sources). NEON sites represent a range of climates, with mean annual temperatures (MAT) ranging from −12 to 25 °C and mean annual precipitation (MAP) ranging from 100 to 2500 mm year$^{-1}$. The sites also capture a variety of ecosystem types, though we focus our analyses primarily on grasslands and forests, which are the most abundant ecosystems across the network. Although it is difficult to leverage observational data collected at a continental spatial scale to probe processes such as priming and mechanisms of SOC persistence that can occur at the micron scale, our work illuminates broad spatial patterns in SOC stocks that may be driven by these underlying processes.

## Results
### Grasslands drive broad-scale fine root and soil carbon relationships
As hypothesized, FRC and SOC were positively related across sites, but these trends were primarily driven by grasslands (Fig. 1b). Combining C stocks across the entire soil profile (to a maximum 2-m depth; Table S2), we found that FRC and SOC were positively related across our continental USA spatial gradient. The total SOC stock in a soil profile was positively related to whole-profile FRC (Fig. S1a; adj $r^2 = 0.39$, $p < 0.001$, $n = 43$) and was best predicted by FRC, MAT, clay content, and ecosystem type (Table S3). Across ecosystem types, for a 1 kg m$^{-2}$ increase in FRC, there was a 7 ± 3 kg m$^{-2}$ increase in SOC (Fig. S1a). Within grasslands, for a 1 kg m$^{-2}$ increase in FRC, there was a 15 ± 2 kg m$^{-2}$ ($p < 0.0001$ in a linear regression)

increase in SOC. This increased to a 23 ± 7 kg m$^{-2}$ increase in SOC if the two highest FRC outliers were excluded ($p = 0.0114$). The grassland-only slope estimate is more than double the value across all ecosystems (cross-ecosystem estimate: 7 ± 3 kg SOC m$^{-2}$ per kg FRC m$^{-2}$; Fig. S1) due to the lack of a FRC-SOC relationship in forests. Our analyses suggest that aridity (MAP standardized by MAT; see "Methods"), micronutrients, and aboveground litter input may be important in explaining forest SOC, but the relationships were not statistically significant (Fig. S2).

When separated into organic and mineral horizons, FRC and SOC remained positively related across ecosystems, and ecosystem type was a significant predictor across most statistical models (Table S4a, b). However, the slope and best predictors of the relationship differed between soil horizons (Fig. S1b). The only significant predictor of SOC in the organic horizon was FRC (adj $r^2 = 0.41$, $p = 0.03$, $n = 17$ out of which 2 were grasslands and rest were forests; Table S4a). Conversely, in the mineral horizon other factors such as MAT and percent clay were also important (adj $r^2 = 0.30$, $p = 0.003$, $n = 43$; Table S4b). Interestingly, three high latitude sites had more than twice as much root biomass than the others: NEON site codes WREF (cold and wet coniferous forest), BARR (tundra), and HEAL (tundra) (See Table S2 for details corresponding to site codes). Excluding these three sites, a model with root biomass, MAT, clay, and ecosystem type still resulted in significant relationships, albeit weaker (adj $r^2 = 0.19$, $p < 0.001$, $n = 40$). Thus, contrary to our hypotheses, our cross-ecosystem analysis suggests that in organic horizons, FRC is a primary predictor of SOC, while in mineral horizons, MAT and clay content are also important. MAT likely is a proxy for temperature limitations on plant productivity and decomposition of SOM, while clay content represents the potential for mineral-associated organic matter formation.

### Strong positive FRC-SOC relationships in deep grassland soils
Similar to the relationships between total FRC and SOC summed across the soil profile, we found that depth distributions of FRC and SOC stocks, quantified using an exponential decay function fit, were related in grasslands but not in forests (Figs. S3–S5, Tables S5a and S5b). Furthermore, we investigated how shallow (<30 cm soil depth) and deep (>30 cm soil depth) FRC influence shallow and deep SOC. Specifically, we tested whether the slope of the standardized FRC-SOC relationship in deep soil layers is higher than the slope of the FRC-SOC relationship in shallow layers. This observation would suggest a more effective SOC accrual per unit FRC at depth. We found that indeed in deep soils, SOC increased more with increasing FRC than in shallow soils in grasslands (Figs. S7 and 2b) but not in forests (Figs. S8 and 2a). Thus, our results support the hypothesis that deep FRC

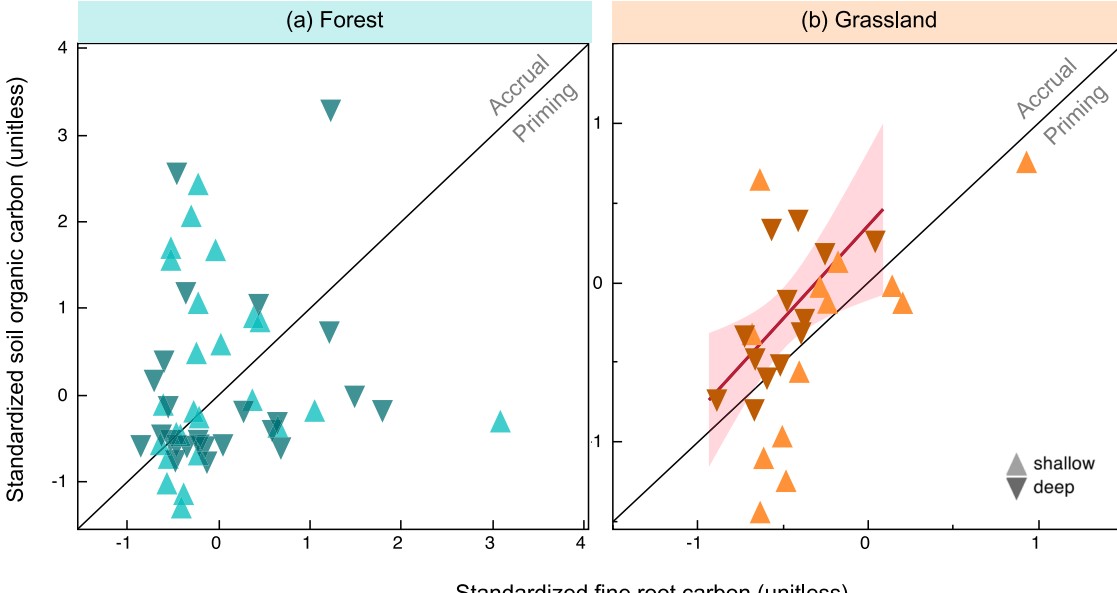

**Fig. 2 | Relationships between standardized fine root carbon and soil organic carbon stocks.** FRC and SOC are shown by soil depth along with a 1:1 line. Data points above the 1:1 line suggest inferred net accrual of SOC while data points below the 1:1 line suggest inferred priming. **a** Forests have no significant relationship between standardized FRC and SOC while (**b**) in grasslands the slope of the relationship between deep roots and deep SOC (>30 cm) is steeper (linear regression slope = 1.2, $p = 0.015$) than the slope between shallow roots and shallow SOC (<30 cm depth; slope = 0.9, $p = 0.19$). Detailed statistics across depths and ecosystem type are provided in Figs. S7 and S8.

increases deep SOC more than shallow FRC increases shallow SOC, but only in grasslands. Furthermore, in grasslands, while a unit increase in standardized deep FRC increases deep SOC by 1.23 ± 0.42 ($p = 0.015$), a unit increase in shallow FRC does not significantly increase shallow SOC ($p = 0.19$) (based on the slopes in Fig. S7).

### Lowest inferred priming in deep grassland soils

Using the standardized shallow and deep FRC-SOC relationships above, we quantified residuals from the 1:1 line, i.e., the difference between observed SOC value at a given site and the expected SOC value at a theoretical 1:1 line. We used these residuals as indicators of net SOC accrual or net priming relative to FRC inputs. Inferred SOC accrual corresponds to relatively more SOC being stored than incoming FRC (i.e., data points above the 1:1 line), and inferred priming corresponds to lower SOC being stored than incoming FRC (i.e., below the 1:1 line) (Fig. 2). We found that forests exhibited great variation in inferred SOC accrual or priming relative to grasslands, which were primarily SOC accruing (Fig. 3). In addition to ecosystem type, the degree of accrual or priming was predicted by factors related to moisture availability (aridity and MAP), soil texture and micronutrients (Table S6). In grasslands, as per our hypothesis, SOC accrual increased with increasing moisture availability, clay content and micronutrients; particularly in shallow soil layers (Fig. S9). In forests, the variability in inferred priming was harder to explain, with the best-fit model explaining up to 44% of the variability in priming versus 73% in grasslands (Table S6). Deep forest soil dynamics remain particularly elusive as we could only explain up to 26% of the variability in inferred priming (Table S6). For example, we observed some indication of higher priming in warmer forests (Fig. S10) compared to cooler forests, but this explained only 20% of the variability in priming.

### Discussion

We found that grasslands drive the relationship between fine root and soil carbon stocks across a continental-scale observational gradient. Our hypotheses about the relationship between FRC and SOC were supported in grasslands but not in forests. Grassland FRC and SOC were strongly positively related, and SOC accrual was highest in high moisture and clay-rich grassland soils. Deep grassland soils had a particularly strong FRC-SOC relationship wherein net SOC accrual was more prevalent than priming. Conversely, forests showed high variability in FRC-SOC relationships. There are several possible explanations for why forest FRC-SOC relationships and priming may be highly variable compared to grasslands. In this section, we use existing literature to discuss possible mechanisms behind our observed differences in grasslands and forests.

### Higher belowground and absorptive root allocation in grasslands relative to forests

Grasslands typically have higher root:shoot ratios than forests and thus grassland root inputs may represent a dominant source of fresh carbon input into the soil[19,31,32]. Conversely, in forests, root:shoot can vary considerably across climate gradients and aboveground inputs may also be an important source of fresh carbon inputs into the soil, especially in deciduous forests[31]. This could mean that FRC inputs and FRC-induced priming are less influential on soil C processes compared to aboveground litter inputs in forests versus grasslands, although we did not see any evidence for aboveground litterfall rates predicting forest SOC either (Fig. S2). It is possible that increased aboveground litterfall also led to priming in forests, as previously observed in temperate and tropical forests[33–36].

Furthermore, relative to forests, grassland plant roots typically have higher absorptive:transport root ratios and a greater proportion of absorptive roots with short lifespans, which could imply greater exudation and greater root litter contribution to SOC, respectively[37,38]. Given the generally greater physiological activities of fine compared to coarse roots[37], a greater relative abundance of high-turnover, absorptive roots in grasslands compared to forests[39] might also result in greater activity rates of soil microbes. This greater microbial activity could also be higher in grasslands than in forests because grassland roots decompose faster than forest roots[40]. Thus, we might expect greater rates of microbial necromass production where fine root abundance is relatively greater, given fine root exudate activities and associated rhizosphere microbial growth and death[41]. Evidence is accumulating that microbial necromass is a meaningful component of persistent SOC stocks[41–43]. We might expect, then, that where absorptive root litter production is greater, necromass and thus SOC accrual might also be greater.

**Fig. 3 | Distributions of inferred soil organic carbon accrual or priming.** SOC accrual or priming is inferred from residuals of the 1:1 line between standardized FRC and SOC; see Fig. 2. Data distributions are shown for shallow (<30 cm depth) and deep (30–200 cm) layers from both (**a**) forest and (**b**) grassland sites. Bins above the zero line are reflective of sites net SOC accrual and bins below the zero line reflect net priming. See Fig. S12 for raw data points of the distributions.

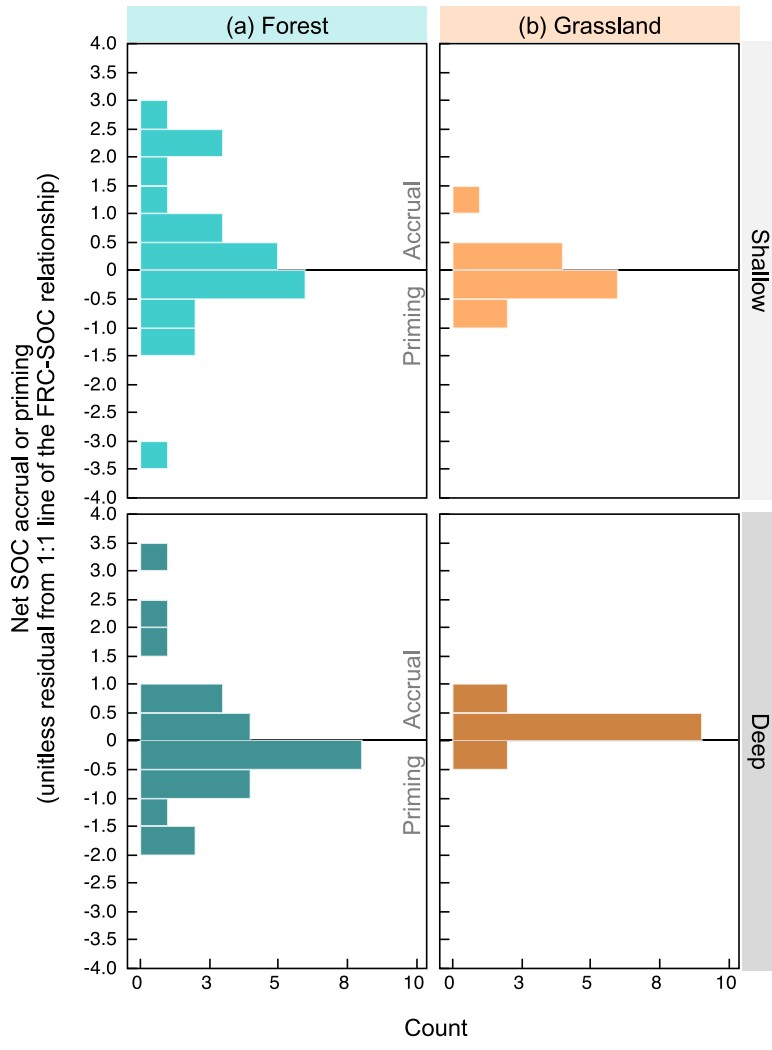

### Variable depth distribution and chemistry of forest roots

In forests and in woody plants, maximum rooting depths vary widely and differ from the more conserved rooting depths of grasslands[18,44]. This rooting depth variation would also influence the relationship between FRC and SOC, given the hypothesis that deeper roots have a greater propensity for mineral interactions and thus contribute to persistent SOC pools[30]. In our data, we also observed generally deeper and more variable depth distributions of FRC and SOC in forests compared to grasslands (Fig. S4). The variability in forest FRC-SOC relationships suggests that factors other than FRC contribute to forest SOC accrual to a greater extent than in grassland soils.

The chemical composition of litter and root inputs may vary more across forest types compared to grasslands, making the relationship between FRC inputs and SOC accumulation across forests harder to predict. Graminoid species can be chemically relatively simple compared to other plants[45]. In contrast, tree species are known to differentially affect decomposition and soil C and N cycling[46,47], including the magnitude of the rhizosphere priming effect[48]. Leaf and root litter from some tree species can be a source of tannins and other phenolics that affect soil processes[49–51], and the presence or absence of tannin-rich species on our forest sites may represent an important source of variability in forest SOC dynamics. For example, tannin-rich litter could promote SOC stabilization while cellulose-rich litter could stimulate priming[52]. Thus, plant litter quality influences priming or stabilization of SOC, and greater variability in litter quality in forests compared to grasslands could potentially contribute to our observed high variation in forest SOC dynamics[53].

### The role of mineralogical limits and moisture availability differs between forests and grasslands

Higher and more variable precipitation in forests compared to grasslands could drive variability in the activity of decomposition enzymes and reactive metals[21]. Thus, mineralogical limits to C storage, driven by variations in the amount and reactivity of clay minerals and reactive metals, may be more important controls on forest SOC than the quantity of plant litter inputs[21,54]. Since our predictive models do not include detailed proxies defining mineralogical limits to C storage, we may be missing some explanatory power and thus seeing high variability in forests. Furthermore, our grassland sites, on average, have half the MAP of our forest sites. Thus, plant growth, litter inputs, and decomposition could all be more moisture-limited in grasslands than in forests, such that C inputs and not mineralogical limits may be a more important factor for stabilizing SOC in grasslands. While we did see support for our hypothesis that moisture availability and SOC accrual are positively related within grasslands (Fig. S9), we did not find support for this hypothesis across ecosystem types.

### Mycorrhizal community complexity may be higher in forests than in grasslands

Lastly, forests have the added complexity of a variety of mycorrhizal symbionts that could be influencing plant-soil processes and belowground carbon allocation in ways different than in grasslands, which are often limited to arbuscular mycorrhizal types that have a lower carbon demand (Fig. S11)[55–57]. Dominant mycorrhizal association type was not a significant predictor in any of our models (see "Methods"), but it is likely that fungal

biomass could have been a better predictor (data unavailable for the NEON megapit samples). FRC and fungal biomass together would better capture the forest variability in total belowground carbon allocation. Furthermore, high variation in mycorrhizal types in forests would also lead to diverse decomposition dynamics[e.g.,58].

Overall, it makes sense that while we were able to explain inferred priming in grasslands using simple climate, soil texture, and nutrient information, inferred priming in forests would require additional predictors, including information about root systems, litter chemistry, mineralogical limits to C storage, and mycorrhizal community structure and function.

## Limitations

Our inferred priming proxy allows us to explore potential relationships between FRC and SOC and offers hypotheses to probe the drivers of SOC formation and persistence. However, our approach has limitations. First, our proxy does not account for all the belowground C inputs into SOC. We only consider standing fine root biomass in our calculations. Ideally, inputs to SOC would include all rhizodeposits and incorporate information about root and fungal turnover rates[59], but these data were unavailable for NEON megapits. Second, we did not have a way to account for variation in the decomposition of incoming root litter. In other words, at sites where we saw high inferred priming, there may have simply been efficient decomposition of fresh root litter. Root litter decomposition rates in the first year can vary widely in forests, with estimates suggesting 20–40% mass loss[60]. Furthermore, turnover times of forest SOC are also more variable than those of grasslands[61]. Thus, we expect that this variability further contributed to the lack of a clear FRC-SOC relationship in forests. Third, as is the case with many observational studies, it is difficult to ascribe causation to correlative relationships. A variety of climate and edaphic factors could be driving both FRC and SOC, thus resulting in the observed relationships.

## Conclusions

In the last few decades, paradigm shifts in SOC research suggest that root carbon inputs are central to organic matter formation and stabilization[5]. We found that at broad spatial scales, FRC and SOC are related, but these trends were driven by grasslands and not forests across a continental-scale gradient. We also investigated whether deeper roots are associated with higher deep SOC, which is presumed to be more stable, and found support for this in grasslands but not in forests. Our hypothesis that stabilization of FRC into SOC would be highest under conditions of high moisture and clay content was also supported in grasslands. Future data collection efforts at the continental scale and beyond should quantify other belowground carbon inputs, such as root turnover and exudation, and microbial biomass turnover to enable a more mechanistic understanding of FRC and SOC linkages across biomes. Nevertheless, in the context of management strategies focused on increasing FRC to increase SOC, root biomass will likely be the main trait that can be measured and managed, and not root turnover[2]. Thus, our analysis provides a useful benchmark of how FRC and SOC are related across broad scales and ecosystems.

Unlike grasslands, forests displayed high variability in FRC-SOC relationships, both across space and depth, likely due to the variability in root-soil interactions across forests. Thus, some forest types may not be ideal settings to increase SOC through fresh root carbon inputs, as this may result in priming-induced SOC losses. Predictors of this forest priming effect, especially in deeper layers, remain elusive and can serve as an important future research trajectory. Conversely, grassland soils with relatively high moisture and clay content may serve as settings in which increased root-derived SOC sequestration may be promoted at depth, although replication across climate gradients and detailed measurements of root dynamics are needed to confidently project SOC accrual.

## Methods
### Sites and data
The root and soil carbon data used for this study were collected by NEON, a continental-scale ecological monitoring program spanning 47 terrestrial sites and all major US ecoclimatic regions. We used data from the NEON "megapits", which include measurements of soil chemical properties, physical properties, and root biomass (Table S1). We downloaded these data from the Soils Data Harmonization (SoDaH) database[24] (data accessed Jul 2020), which includes NEON data[62] among other network data sources. Data from other networks were not considered in this study because sites infrequently measured root biomass and soil chemistry profiles to depth in the same location. Four NEON sites (STER, KONA, PUUM, and TOOL) were excluded from our study because root data were not collected in the NEON megapits. Thus, 43 of the 47 NEON sites were included in our analyses. All 43 sites had mineral soil horizons, and 17 of the 43 sites had organic soil horizons (15 forests and 2 high-latitude grasslands). Site metadata such as MAT, MAP, and site-wide dominant plants were taken from SanClements et al.[63]. We also calculated an aridity index as MAP (mm) standardized by MAT (°C) using the formula MAP/(MAT + 13). The 13 was added to adjust for negative MAT values[64]. Lower values represent more arid conditions. Land cover was ascertained by NEON scientists and was based on the NLCD land cover classifications[65] from NEON field site information tables (link). In the manuscript text, we refer to land cover as "ecosystem type".

### Sample collection and processing
The NEON megapit sampling effort was a one-time measurement conducted by NEON staff and the USDA Natural Resource Conservation Service (NRCS) over the course of 2014–2018. At each site, NEON scientists and contractors excavated a 2 m deep (or to bedrock) soil pit in the vicinity of the NEON eddy covariance tower. The timing of sampling varied across the growing season and was not always at peak biomass. NRCS soil scientists then assigned soil taxonomy in situ, and by taxonomic horizon to the bottom of the pit. These samples were then sent to the Kellogg Soil Survey Laboratory in Lincoln, Nebraska where, after passing through a 2 mm sieve, they were analyzed for a host of physical and chemical properties including bulk density, particle size, total C, nitrogen (N), phosphorous (P), metals, and other edaphic properties using standard NRCS methods[66].

At each NEON megapit, root samples were collected across depth profiles. Samples were collected in 10-cm depth increments to 1 m depth, then in 20-cm depth increments to 2 m depth by cutting 10-cm deep × 10 cm wide soil monoliths, in three vertical profiles on the left, middle, and right side of the pit. Roots were hand-sorted from these monoliths, visually classified as live or dead, and the diameter was measured. Most NEON sites (30 sites out of 43) classified "fine roots" as less than 2 mm in diameter. However, 13 sites had a different methodological protocol and used a 4 mm diameter cutoff. Of these, nine were forests, three were rangelands/grasslands (hereafter, grassland), and one was a shrubland. Note that we statistically tested the hypothesis that having different fine root diameter cut offs may be related to our observed variability in forest FRC-SOC relationships. This hypothesis was not supported in a Wilcoxon rank test of the residuals from the FRC-SOC relationship when comparing the two diameter classes ($Z = 0.59$, $p = 0.55$). Other studies[16] have also found that root biomass from these two diameter classes are highly correlated across sites in fine root databases, and that in the NEON sites, the diameter sampling differences do not influence properties such as rooting depth distribution[16].

Root biomass was measured after oven-drying the samples at 65 °C for at least 48 h. Dried root samples were sent to the University of Wyoming for analysis of C concentrations using elemental analysis. The three vertical pit profiles per megapit were averaged prior to ingestion into SoDaH and used in our statistical analysis. Despite the 4 mm diameter exceptions, we consider the rootstock to represent "fine-root biomass" throughout the manuscript.

Lastly, we used annual litterfall fluxes (forest sites only) as a covariate in an exploratory analysis (see Table S1 for data sources). Briefly, annual litterfall was measured by collecting all material dropped from the forest canopy with a diameter <2 cm and a length <50 cm using elevated 0.5 m$^2$ PVC traps. Traps were deployed (20 plots per site) near the megapits. We used the total mass (leaves, needles, twigs, etc. all added) collected by the

traps over the course of a growing season to estimate annual productivity. Where multiple years of data were available, the average flux was used.

## Data alignment

Alignment of FRC and SOC data was necessary due to different sampling strategies for roots and soils. Roots were sampled at fixed (10 or 20 cm) increments through the profile, while soils were sampled once in each taxonomic horizon regardless of horizon depth. Therefore, we aligned the root data with the corresponding soil horizon. Fine root biomass C stocks (FRC) were calculated as the product of root biomass (g m$^{-2}$) in a given depth interval and FRC concentration (%). SOC stocks were calculated as the product of soil organic C concentration (%), bulk density (kg/cm$^3$), and sampling depth (cm), then converted to kg m$^{-2}$ by multiplying by $1 \times 103$.

## Calculation of beta coefficients

In order to investigate the relationship between depth profiles of FRC and SOC, we calculated beta values using an exponential decay curve (Eq. 1), which describes how stocks change with depth[20,67]. Of the 43 NEON sites, 36 were used to calculate beta coefficients. Seven sites (BARR, CLBJ, GRSM, GUAN, JORN, LAJA, TEAK) were excluded because beta coefficients could not be calculated due to too few SOC measurements in the profile. SOC and FRC may accumulate primarily in surface soils and to varying degrees deeper in the profile, or there may be a gradual and consistent increase at each depth interval. These different accumulation patterns can be captured in an exponential function (Eq. 1), where a higher beta coefficient indicates a deeper distribution of root or SOC, relative to a lower beta. We converted each depth profile (FRC or SOC) at each site into one beta coefficient to facilitate these analyses.

Beta coefficients ($\beta$) were calculated using the following function[20,67]:

$$Y = 1 - (\beta)^d \qquad (1)$$

In Eq. 1, $Y$ is the cumulative fraction of either SOC or root biomass in a given layer with respect to the whole profile, and $d$ is the depth (cm) measured at the bottom of that layer. For every depth layer at every site, we solved for $\beta$ in Eq. 1 for both SOC ($\beta_{SOC}$) and root biomass ($\beta_{roots}$). These $\beta$ values were used as starting parameters at discrete points through the depth profile, and we used the iterative Bound Approximation by Quadratic Approximation (BOBYQA) method (package minqa in R Statistical Software)[68] to interpolate between points and resolve the function across the continuous depth profile at each site[69]. The BOBYQA-resolved $\beta_{SOC}$ and $\beta_{FRC}$ values were used as response and fixed effect variables, respectively, in mixed effects models.

## Statistical analyses

We analyzed relationships between SOC, FRC, and other climatic and edaphic covariates using linear mixed-effects models. We analyzed FRC-SOC relationships in three ways wherein FRC and SOC stocks were: (1) summed across the whole profile, (2) separated by organic and mineral soil horizons, and (3) described as beta coefficients as a function of depth. For each analysis, we constructed a null (random effects only) model, a full model, and then reduced models that lacked covariates. We selected best-fit models based on the lowest Akaike Information Criterion (AIC) score aside from the full model, to avoid overparameterization, or the sample-size corrected AIC (AICc) score when the data set contained fewer than 40 observations[70]. In whole-profile and by-horizon analyses, the full model included SOC as the response and FRC, MAT, MAP, clay percent, and land cover (ecosystem type) as fixed effects and maximum profile depth as a random effect. We also verified that sampling depth does not influence our analyses by conducting a multiple regression analysis including all covariates and maximum profile depth and found that profile depth was not a significant parameter ($p = 0.57$, partial $r^2 = 0.01$). Lastly, we explored the role of mycorrhizal associations by adding dominant mycorrhizal type (arbuscular, ectomycorrhizal or mixed prescribed based on NEON reported species)[55] as a fixed effect but saw no significant relationships or model improvements. Across the mixed-effects models, we report the significance

level ($p$-value) calculated using Satterthwaite's method (lmerTest R package)[71,72], a test statistic ($\chi^2$), and marginal pseudo-$R^2$ (sjstats R package)[73]. The fixed effects of the best fit model were tested using analysis of variance (Anova function in the R package car)[74]. Forest and grassland land cover types were tested and shrublands and cultivated lands were excluded from this analysis due to limited sample size. Assumptions of homoscedasticity, low variance inflation factors and normal data distributions were verified for each statistical model. R code is available in Supplementary Data 1.

## Inferred SOC accrual and priming

We calculated inferred net accrual or priming using residuals, which measure the difference between observed and predicted values, from the shallow (<30 cm) and deep (>30 cm depth) FRC-SOC relationships. Specifically, we determined the residual difference between the observed SOC and expected SOC along a standardized 1:1 line, which represents a theoretical scenario where each unit of FRC input results in an equivalent unit increase in SOC (Figs. S7 and S8). This 1:1 line serves as the baseline for assessing if SOC levels are higher or lower than expected based on FRC inputs. The calculated residuals, therefore, act as a proxy for SOC accrual (when observed values exceed the expected) or priming (when observed values are less than expected) in relation to FRC inputs. Furthermore, we employed multiple regression models to explore the potential factors influencing this inferred priming or accrual, as detailed in Table S6. These models allow us to identify and evaluate variables that may affect the relationship between FRC and SOC across different sites and conditions.

## Reporting summary

Further information on research design is available in the Nature Portfolio Reporting Summary linked to this article.

## Data availability

All data used in this manuscript are available in a public repository in the following National Ecological Observatory Network (NEON) data products: Soil physical and chemical properties, Megapit (DP1.00096.001, accessed January 1, 2020); Root biomass and chemistry, Megapit (DP1.10096.001, accessed January 1, 2020); and Litterfall and fine woody debris production and chemistry (DP1.10033.001, accessed January 1 2021); PROVISIONAL. Data can be accessed at https://data.neonscience.org/.

## Code availability

The code used in the study is available as an R source file in Supplementary Data 1.

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

## Acknowledgements

A.M. was supported by the Gordon and Betty Moore Foundation (Grant GBMF5439), the Laboratory Directed Research and Development (LDRD) Program at Pacific Northwest National Laboratory under U.S. Department of Energy (DOE) contract DE-AC05-76RL01830, and the Swiss National Science Foundation (project 200021_215214). This work was also supported by the Long Term Ecological Research Network Office (LNO; NSF awards 1545288 and 1929393) and the National Center for Ecological Analysis and Synthesis at the University of California Santa Barbara (Awarded to K.L. and W.R.W.). The National Ecological Observatory Network (NEON) is a program sponsored by the U.S. National Science Foundation and operated under cooperative agreement by Battelle. This material is based in part upon work supported by the U.S. National Science Foundation through the NEON Program. K.G. was supported by the U.S. DOE "Microbes Persist" Scientific Focus Area (SCW1632) at LLNL, under the auspices of DOE Contract DE-AC52-07NA27344. SJTD acknowledges support from the US National Science Foundation (NSF-DEB-LTER 1832210). W.R.W. acknowledges support from NSF award numbers (2224439 and 1926413).

## Author contributions

A.M., J.A.M., S.W-L., K.G., E.K-S., A.S.G., D.P., K.L., W.R.W., R.B.J., S.J.T-D. and M.L. conceptualized the study. A.M., J.A.M., S.W-L. and K.G. performed the analyses. A.A.B., S.A.B., M.G., J.M.F., A.S.G., C.M., D.P., K.L., W.R.W., and S.W-L. provided data. AM wrote the manuscript with input from all coauthors. R.B.J., K.L. and W.R.W. acquired funding.

## Competing interests

The authors declare no competing interests.

## Additional information

[1]Department of Earth System Science, Stanford University, Stanford, CA, USA. [2]Biological Sciences Division, Pacific Northwest National Laboratory, Richland, WA, USA. [3]Department of Geography, University of Zürich, Zürich, Switzerland. [4]Microbiology Department, University of Tennessee, Knoxville, TN, USA. [5]National Ecological Observatory Network, Battelle, Boulder, CO, USA. [6]Department of Biological and Ecological Engineering, Oregon State University, Corvallis, OR, USA. [7]Physical and Life Sciences Directorate, Lawrence Livermore National Laboratory, Livermore, CA, USA. [8]Department of Life and Environmental Sciences; University of California, Merced, Merced, CA, USA. [9]Department of Ecology and Evolutionary Biology and Kansas Biological Survey & Center for Ecological Research, University of Kansas, Lawrence, KS, USA. [10]Department of Biological Sciences, Boise State University, Boise, ID, USA. [11]Department of Natural Resources and Environmental Sciences, University of Illinois, Urbana, IL, USA. [12]Department of Natural Resources and the Environment, University of New Hampshire, Durham, NH, USA. [13]Carbon Direct, New York, NY, USA. [14]New York University, New York, NY, USA. [15]Rocky Mountain Research Station, United States Forest Service, Boise, ID, USA. [16]Department of Environmental Studies, Dartmouth College, Hanover, NH, USA. [17]Department of Crop and Soil Science, Oregon State University, Corvallis, OR, USA. [18]Climate and Global Dynamics Laboratory, National Center for Atmospheric Research, Boulder, CO, USA & Institute of Arctic and Alpine Research, University of Colorado, Boulder, CO, USA. [19]Woods Institute for the Environment and Precourt Institute for Energy, Stanford University, Stanford, CA, USA.
✉e-mail: avni.malhotra@pnnl.gov

