## [Transparent Peer Review file · Communications Earth & Environment]

Fine root and soil carbon stocks are positively related in grasslands but not in forests

Corresponding Author: Dr Avni Malhotra

Version 0:

Reviewer comments:

Reviewer #1

(Remarks to the Author)

This work provides new insight into the relationship between fine root carbon and soil organic carbon in light of SOC accrual and priming effect. The approach used to indicate the relative contributions of these two processes requires stronger justification. Additionally, some discussions are speculative, particularly those regarding the root-to-shoot ratio, mineralogical limits, and chemical composition of litter and roots. My specific comments are provided in the attached PDF file.

Reviewer #2

(Remarks to the Author)

The authors address the suggestion that fine roots are an important source of SOC by establishing the relationship between fine root biomass carbon and SOC across a wide range of ecosystems in USA. They show a strong positive relationship between the two variables in grassland ecosystems but a variable relationship in forests. The study is a useful contribution to the topic, as it tests one line of evidence that probes the relative contribution of roots to SOC, and come up with compelling evidence of a relationship in grasslands, and the environmental factors that also influence SOC. The issue with this sort of study is that it is not possible to determine which factors are driving SOC - is it roots directly or is it that the environmental factors drive both SOC and root biomass C independently? This issue needs to be explicitly addressed in the Discussion, and also needs to be recognized throughout the Results and Discussion by avoiding statements such as:

"our inferred priming proxy allows us to isolate the influence of FRC on SOC"

"in grasslands deep roots have a larger effect on deep SOC than shallow roots"

Such statements infer that it has been possible to isolate cause and effect, which is not possible in a study of this design.

It might also be worth exploring if the variability in fine root diameter used at the forest sites contributed to the variable relationships found at forested sites.

All other caveats I thought of were dealt with well in the Discussion.

The manuscript is clearly and efficiently written.

Reviewer #3

(Remarks to the Author)

This manuscript, titled "Continental-scale relationships of root and soil carbon hold in grasslands but not forests," is a valuable contribution to understanding root-soil carbon dynamics across ecosystems. The title is clear but could benefit from a more concise or specific phrasing to immediately capture the essence of the findings. Notably, there is a lack of context introduction about grassland and forest ecosystems, particularly regarding their root dynamics and carbon sequestration potential, as the title suggests. This omission is critical because much of the discussion heavily contrasts the differences in fine root carbon (FRC) and soil organic carbon (SOC) relationships between these two ecosystems. While the contrasting mechanisms in grasslands and forests are intriguing, they require clearer articulation and stronger integration into the narrative. Given the significant and valuable theme, I recommend revising the manuscript before publication. Below are detailed suggestions for improving the manuscript:

Introduction

The introduction provides useful context but could be improved by sharpening the hypotheses. Specifically, explicitly differentiate how grasslands and forests are expected to vary in their root-soil carbon relationships to set up the study's

objectives more effectively.

Results and Discussion

Authors are encouraged to enrich this section by adding and complementing citation evidence to support or compare the results with existing studies. For example, some paragraphs lack references and contain only descriptive results without in-depth discussion. Quantitative comparisons between grassland and forest systems would further strengthen the results. Additionally, the discussion on priming mechanisms in forests lacks clarity; more context and relevant citations should be provided to explain the observed variability. Expanding and deepening this section would significantly enhance the quality of the paper.

Conclusions

The conclusion needs to be better integrated, summarized, and refined, particularly by emphasizing the most important findings of the study and their implications. Including recommendations based on the results would further increase the impact of this section.

Materials and Methods

This section should be revised concisely and polished for better readability. For example, the phrase "low sample size" (L410) could be revised to "limited sample size" or "small sample size" for improved clarity and professionalism.

References

The reference format should be checked and revised for consistency. Ensure that all citations are accurate, uniformly formatted, and correctly aligned with the journal's guidelines.

In general, this manuscript addresses an important and timely topic in the field of ecosystem carbon dynamics. With the suggested revisions, the clarity, depth, and overall quality of the manuscript will be substantially improved, increasing its potential impact and suitability for publication.

Version 1:

Reviewer comments:

Reviewer #1

(Remarks to the Author)

This is a substantially revised manuscript. From my perspective, the authors have mostly addressed all of my previous comments. I have no further remarks at this time.

Reviewer #2

(Remarks to the Author)

The authors have satisfactorily addressed all of my concerns.

Reviewer #3

(Remarks to the Author)

This is a well-executed, conceptually rich, and increasingly polished manuscript. The revisions have successfully addressed most of the key issues from the previous round. I recommend acceptance after minor revision, mainly for clarity and consistency.

Specifically, the Results section emphasizes grasslands in all subtitles, while the Discussion includes only one forest-specific section without referring back to grasslands in the subtitle. I suggest improving the structural correspondence between Results and Discussion.

In addition, please ensure consistency across the Abstract, Results, and Discussion, and refine the Conclusion to better highlight key insights and broader implications.

Thank you to the authors for their thoughtful revisions.

Please find author reviewer responses below. Original reviewer comments are in blue. Our responses are in black. Any new text added into the manuscript is in red and can also be seen in the tracked changes file.

Reviewer #1

This work provides new insight into the relationship between fine root carbon and soil organic carbon in light of SOC accrual and priming effect. The approach used to indicate the relative contributions of these two processes requires stronger justification. Additionally, some discussions are speculative, particularly those regarding the root-to-shoot ratio, mineralogical limits, and chemical composition of litter and roots. My specific comments are provided in the attached PDF file.

We thank the reviewer for their detailed feedback. We respond to specific comments from the PDF below.

L53: what do you mean by inferred priming

Response: We now delete this term and better explain our analysis later in the abstract. The new text now reads:

Here, we tested the relationship of SOC and FRC using data from 43 sites across the US National Ecological Observatory Network (NEON). We found that total stocks of SOC and FRC in the top 2 meters of soil were positively related with an across-ecosystem slope of 7 ± 3 kg SOC m⁻² per kg FRC m⁻². However, grassland sites primarily drove this relationship. Grasslands had 15 ± 2 kg SOC m⁻² per kg FRC m⁻², which is double the across-ecosystem slope. We used deviations from the standardized 1:1 relationship between FRC and SOC to infer whether ecosystems were net priming (indicated by observed SOC being lower than the 1:1 line) or SOC accruing (higher SOC than the 1:1 line).

L94: It should be clearly explained why the inference derived from 1:1 relationship is valid, as readers may find it difficult to understand how SOC accrual and priming were inferred.

Response: Thanks for flagging this. We have reframed the paper slightly, wherein we first and foremost emphasize the relationship between FRC and SOC. The net SOC accrual vs priming inference comes later and we edited text in the abstract (see comment above) and objectives to explain the approach more clearly. We also added a methods section focused on this approach (detailed in a later reviewer comment). Revised objectives paragraph is as follows:

Here, we used a natural gradient with varying fine root biomass carbon stocks (hereafter, FRC) to explore the relationship between FRC and SOC stocks. We also tested how the FRC-SOC relationship varies by ecosystem type, soil depth and soil horizon (organic or mineral), and how the relationship is influenced by climate, mineralogy, and soil nutrients. We expected grasslands to have stronger FRC-SOC relationships than forests because of the high below:aboveground

biomass ratio in grasslands^{17,18}. We also expected that mineral horizons would have a stronger relationship between FRC and SOC than organic horizons because the latter are likely more influenced by aboveground litter inputs⁵. Additionally, in mineral horizons where SOC stabilization can proceed via organo-mineral interactions, FRC is likely linked to net SOC to a greater extent than in organic horizons, where SOC may be less protected from microbial attack and microbial population abundances tend to be higher⁵.

Furthermore, by comparing our observed FRC-SOC relationships with a theoretical one-to-one relationship between FRC input and SOC stock, we inferred net SOC accrual versus priming across the gradient. Specifically, we assumed that sites with observed SOC above the 1:1 line of standardized SOC and FRC data indicate the potential for net SOC accrual (hereafter, SOC accrual), whereas sites with SOC below the 1:1 FRC-SOC line indicate the potential for net priming (hereafter, priming). We hypothesized that SOC accrual would be highest in ecosystems with high moisture and clay content where plant production and mineral stabilization of fine root litter would be optimized²⁵. Conversely, priming would be more likely in ecosystems with lower moisture and clay content, where SOC would have a lower probability of interacting with soil minerals. We also hypothesized that SOC accrual would be greater at depth due to higher concentrations of reactive minerals and/or metals and lower microbial abundance/activity than in surface soils²⁶.

L101: This point about higher concentrations of reactive minerals and metals in the subsurface should be explained or introduced earlier.

Response: Good point, thanks. We now introduce this point in previous (second introduction) paragraph as follows:

Whether fine root carbon inputs drive SOC accrual or priming is expected to vary by ecosystem and vegetation type, soil moisture, SOC stock and its distribution between particulate and mineral-associated pools, the amount and reactivity of soil minerals, and macro and micro soil nutrients^{8,16,17}. Since these factors vary throughout the organic and mineral layers of soils, we also expect soil depth and horizons to be important predictors of the relationship between fine root carbon and SOC.

L128: But does the stronger role of grassland roots also depend on the decomposability of root litter, i.e. higher inputs combined with greater decomposition contributing to higher SOC?

Response: Reviewer raises a good point as do other reviewers about further expanding the discussion on forests vs grasslands. We have removed this paragraph and incorporated it into a new discussion section. We also added information about grassland root decomposability and its influence on SOC. The relevant new discussion paragraph is as follows:

Fifth, grassland plant roots typically have higher absorptive:transport root ratios and a greater proportion of absorptive roots with short lifespans, which could imply greater exudation and greater root litter contribution to SOC, respectively^{42,43}. Given the generally greater physiological

activities of fine compared to coarse roots, a greater relative abundance of high-turnover, absorptive roots in grasslands compared to forests⁴⁴ might also result in greater activity rates of soil microbes. This greater microbial activity could also be higher in grasslands than in forests because grassland roots decompose faster than forest roots⁴⁵. We thus might expect greater rates of microbial necromass production where fine root abundance is relatively greater, given fine root exudate activities and associated rhizosphere microbial growth and death⁴⁶. Evidence is accumulating that microbial necromass is a meaningful component of persistent SOC stocks⁴⁶⁻⁴⁸. We might expect, then, that where absorptive root litter production is greater, necromass and thus SOC accrual might also be greater.

L134: Do you know exactly what the variations in rooting depths are? Without this information, your assumption seems less convincing.

Response: Great point. We do have this information in the SI and now better link the Figure S4 to this point. We also conducted a new analysis and calculated the depth at which 50% or 95% of the root biomass occur in grasslands vs forests.

New text: In our data we also observed deeper and more variable depth distributions in forests compared to grasslands (Figure S4).

New text in S4 caption: Using these root beta values, we also calculated the depth at which 50% or 95% of the root biomass occur. For forests, 50% of the roots occurred above 91 cm and 95% of the roots occurred above 106 cm. Conversely, for grasslands, 50% of the roots occurred above 65 cm and 95% occurred above 76 cm.

L139: So, do all the ecosystems, particularly grasslands, have both organic and mineral horizons? The presence of an O horizon may depend on the rate of decomposition.

Response: No, only 17 sites had an O horizon out of which 2 were grasslands and the rest were forests. Sample size of the mineral versus organic horizon analyses were already provided in the text and in the associated figure S1b caption. But we have added new text in both locations to clarify to what extent grasslands and forests were represented.

New text reads as follows: The only significant predictor of SOC in the organic horizon was FRC (adj $r^2 = 0.41$, $p = 0.03$, $n = 17$ out of which 2 were grasslands and rest were forests; Table S4a)

L139: This separation of soil horizons and the associated differences in the predictors have neither been introduced in the background nor mentioned in the hypotheses.

Response: We agree and have fixed this both in the second paragraph of the introduction where we provide background and in the third paragraph where we provide the hypotheses. New relevant text in background:

Whether fine root carbon inputs drive SOC accrual or priming is expected to vary by ecosystem and vegetation type, soil moisture, SOC stock and its distribution between particulate and mineral-associated pools, the amount and reactivity of soil minerals, and macro and micro soil nutrients^{8,16,17}. Since these factors vary throughout the organic and mineral layers of soils, we also

expect soil depth and horizons to be important predictors of the relationship between fine root carbon and SOC.

New text in hypotheses:

We also expected that mineral horizons would have a stronger relationship between FRC and SOC than organic horizons because the latter are likely more influenced by aboveground litter inputs²⁴. Additionally, in mineral horizons where SOC stabilization can proceed via organo-mineral interactions, FRC is linked to net SOC to a greater extent than in organic horizons, where SOC may be less protected from microbial attack and microbial population abundances tend to be higher⁵.

L146: Please see the comment immediately above. These characters were never foreshadowed in the Introduction or your hypotheses.

Response: If the reviewer meant this regarding mineral and organic horizons, we have fixed this (see last response). In case the reviewer meant this regarding site codes, we also address this since the inclusion of site code names in the text can be confusing. We have edited this sentence to clarify that these are NEON site codes and that their detailed information can be found in Table S2. Revised text:

Interestingly, three high latitude sites had more than twice as much root biomass than the others: NEON site codes WREF (cold and wet coniferous forest), BARR (tundra), and HEAL (tundra) (See Table S2 for details corresponding to site codes).

L164: For almost for all the FRC and SOC mentioned previously, no information was provided regarding whether they are concentrations or stocks.

Response: Thank you for flagging this important point. The information was already in methods but we agree that it is important to clarify this in the main text. We have fixed this in the first mention of FRC and SOC in the introduction.

New introduction text reads:

Here, we used a natural gradient with varying fine root biomass carbon stocks (hereafter, FRC) to test the relationship between FRC and SOC stocks.

L186: This may need further explanation in the M&M section. What do you mean by 'residuals'?

Response: we corrected this in three places in the manuscript:

1) New text at the location of this reviewer comment explains this further as follows:

Using the standardized shallow and deep FRC-SOC relationships above, we quantified residuals from the 1:1 line (i.e., the difference between observed SOC value at a given site and SOC value of at the 1:1 line) and use this as a proxy for priming relative to FRC inputs.

2) We also clarify our approach in the introduction last paragraph (as per previous R1 comment) with new text explaining the approach:

Specifically, we assumed that sites with observed SOC above the 1:1 line of standardized SOC and FRC data indicate the potential for net SOC accrual (hereafter, SOC accrual), whereas sites with SOC below the 1:1 FRC-SOC line indicate the potential for net priming (hereafter, priming).

3) We also added a new section at the end of the methods section to further detail this:

Inferred SOC accrual and priming

We calculated inferred net accrual or priming using residuals, which measure the difference between observed and predicted values, from the shallow (<30 cm) and deep (>30 cm depth) FRC-SOC relationships. Specifically, we determined the residual difference between the observed SOC and expected SOC along a standardized 1:1 line, which represents a theoretical scenario where each unit of FRC input results in an equivalent unit increase in SOC (Figure S7 and S8). This 1:1 line serves as the baseline for assessing if SOC levels are higher or lower than expected based on FRC inputs. The calculated residuals, therefore, act as a proxy for SOC accrual (when observed values exceed the expected) or priming (when observed values are less than expected) in relation to FRC inputs. Furthermore, we employed multiple regression models to explore the potential factors influencing this inferred priming or accrual, as detailed in Table S6. These models allow us to identify and evaluate variables that may affect the relationship between FRC and SOC across different sites and conditions.

L215: This section, primarily a discussion part, is highly speculative.

Response: We agree and our intention was to keep this part speculative. Our goal was to highlight possible reasons (based on current literature) behind the unexplained variation in forest fine root and soil organic carbon stock. Our hope is that this discussion will motivate future research to explain this variability. We have restructured the manuscript to separate results and discussion and now clarify in the beginning of this section that this is a discussion with the following new sentence:

In this section, we use existing literature to discuss possible mechanisms behind our results and hope that these hypotheses will be tested in future research.

We have also added more references into the discussion, ensuring that each point is substantiated by existing literature.

L226: Why do forests show mineralogical limits, whereas grasslands do not? Even if this is the case, how could it explain the higher variability in forests compared to grasslands?

Response: We have clarified this point as follows now in the discussion:

Third, higher and more variable precipitation in forests compared to grasslands could drive variability in the activity of decomposition enzymes and reactive metals¹⁹. Thus, mineralogical limits to C storage, driven by variations in the amount and reactivity of clay minerals and reactive metals, may be more important controls on forest SOC than litter inputs^{19,33}. Since our predictive models do not include detailed proxies defining mineralogical limits to C storage, we may be missing some explanatory power and thus seeing high variability in forests. Furthermore, our grassland sites, on average, have half the mean annual precipitation of our forest sites. Thus, plant

growth, litter inputs and decomposition could all be more moisture limited in grasslands than in forests, such that C inputs and not mineralogical limits are a more important factor for stabilizing SOC in grasslands.

L232: Do you have any evidence?

Response: Unfortunately, we do not have chemical quality data for the roots. Thus, the reviewer is correct in pointing out that we do not have data on this aspect but can only discuss what is known from published literature. In the revised manuscript, we now make it clear that this is a discussion and not a result. The section in which this paragraph is now starts with:

In this section, we use existing literature to discuss possible mechanisms behind our results and hope that these hypotheses will be tested in future research.

L266: The concept of 'space-for-time substitution' appears only at the beginning and at the end of you paper.

Response:

After thinking more about this, we decided that the reviewer is correct in that this concept is tangential to the core message of the paper. Thus, we have removed the mention of space-for-time substitution from the paper.

Reviewer #2

The authors address the suggestion that fine roots are an important source of SOC by establishing the relationship between fine root biomass carbon and SOC across a wide range of ecosystems in USA. They show a strong positive relationship between the two variables in grassland ecosystems but a variable relationship in forests. The study is a useful contribution to the topic, as it tests one line of evidence that probes the relative contribution of roots to SOC, and come up with compelling evidence of a relationship in grasslands, and the environmental factors that also influence SOC. The issue with this sort of study is that it is not possible to determine which factors are driving SOC - is it roots directly or is it that the environmental factors drive both SOC and root biomass C independently? This issue needs to be explicitly addressed in the Discussion,

Response: We thank the reviewer for their encouragement and agree with the caveat they raised. In our new discussion section, we detail a variety of factors that could be leading to our observations (see new Discussion section). We have also added this caveat explicitly into the new Limitations section as follows:

Third, as is the case with many observational studies, it is difficult to ascribe causation to correlative relationships. A variety of climate and edaphic factors could be driving both FRC and also SOC, thus resulting in the observed relationships.

and also needs to be recognized throughout the Results and Discussion by avoiding statements such as:

"our inferred priming proxy allows us to isolate the influence of FRC on SOC"

Response: we have edited this as follows:

Our inferred priming proxy allows us to explore potential relationships between FRC and SOC, and offers a means of developing hypotheses to probe the drivers of SOC formation and persistence.

"in grasslands deep roots have a larger effect on deep SOC than shallow roots"

Response: we have edited this as follows:

in grasslands the slope of the relationship between deep roots and deep SOC (>30 cm) is steeper (linear regression slope= 1.2, p=0.015) than the slope between shallow roots and shallow SOC (<30 cm depth; slope=0.9, p=0.19)

Such statements infer that it has been possible to isolate cause and effect, which is not possible in a study of this design.

Response: We agree and have fixed this throughout.

It might also be worth exploring if the variability in fine root diameter used at the forest sites contributed to the variable relationships found at forested sites.

Response: Reviewer brings up a great point. Following this comment, we conducted a Wilcoxon rank test between the two categories of sampling cut offs, 4 mm (n= 9 forest sites) and 2 mm (n =

16 forest sites). Specifically, we tested the null hypothesis that there is no difference in the residuals of the FRC-SOC relationship among the two diameter classes in forest. This null hypothesis was supported ($Z= 0.59$, $p= 0.55$). Thus, we infer that the variability in fine root diameter cut is unrelated to the variability in the FRC-SOC relationships. We have now added this caveat in the methods section where we mention that NEON used different size cut offs due to sampling logistical constraints. New text reads:

Note that we statistically tested the hypothesis that having different fine root diameter cut offs may be related to our observed variability in forest FRC-SOC relationships. This hypothesis was not supported by a Wilcoxon rank test of the residuals from the FRC-SOC relationship when comparing the two diameter classes ($Z= 0.59$, $p= 0.55$). Other studies⁵⁸ have also found that root biomass from these two diameter classes are highly correlated across sites in fine root databases, and that in the NEON sites, the diameter sampling differences do not influence properties such as rooting depth distribution⁵⁸.

All other caveats I thought of were dealt with well in the Discussion.

The manuscript is clearly and efficiently written.

Thank you

Reviewer #3

This manuscript, titled "Continental-scale relationships of root and soil carbon hold in grasslands but not forests," is a valuable contribution to understanding root-soil carbon dynamics across ecosystems. The title is clear but could benefit from a more concise or specific phrasing to immediately capture the essence of the findings.

Response: Thank you for your encouragement. Also based on other reviewer comments, we have now added more specificity to the title as follows (new words are underlined):

Continental-scale relationships of fine root and soil carbon stocks hold in grasslands but not forests

Notably, there is a lack of context introduction about grassland and forest ecosystems, particularly regarding their root dynamics and carbon sequestration potential, as the title suggests. This omission is critical because much of the discussion heavily contrasts the differences in fine root carbon (FRC) and soil organic carbon (SOC) relationships between these two ecosystems. While the contrasting mechanisms in grasslands and forests are intriguing, they require clearer articulation and stronger integration into the narrative. Given the significant and valuable theme, I recommend revising the manuscript before publication. Below are detailed suggestions for improving the manuscript:

Response: We agree with these recommendations. Other reviewers had the same concerns. Now our new introduction contains specific background and hypotheses about the role of ecosystem type and hypotheses about forests vs. grasslands. We have made each of the suggested changes (detailed below):

Introduction

The introduction provides useful context but could be improved by sharpening the hypotheses. Specifically, explicitly differentiate how grasslands and forests are expected to vary in their root-soil carbon relationships to set up the study's objectives more effectively.

Response: We now do this in the introduction's second and third paragraph. First we set up rationale in the second paragraph:

Whether fine root carbon inputs drive SOC accrual or priming is expected to vary by ecosystem and vegetation type, soil moisture, SOC stock and its distribution between particulate and mineral-associated pools, the amount and reactivity of soil minerals, and macro and micro soil nutrients⁶⁻⁸. Ecosystem types with different dominant vegetation would also vary in their accrual and priming behaviors due to variation in belowground allocation, rooting depth and other root traits¹⁶⁻¹⁸

Next, we detail hypotheses in the third introduction paragraph as follows:

We expected grasslands to have stronger FRC-SOC relationships than forests because of the high below:aboveground biomass ratio in grasslands^{17,18}.

Results and Discussion

Authors are encouraged to enrich this section by adding and complementing citation evidence to support or compare the results with existing studies. For example, some paragraphs lack references and contain only descriptive results without in-depth discussion.

Response: Following this comment and the comments from other reviewers, we have restructured the results and discussion as separate sections now. The discussion now has three new paragraphs on possible reasons for our observed differences between forest and grassland. We have added several new references in this section. We have also streamlined the discussion text from the previous version. New discussion text is as follows (paragraphs not shown contain points from the previous manuscript version):

Second, in forests and in woody plants, maximum rooting depths vary widely and differ from the more conserved rooting depths of grasslands^{16,32}. This rooting depth variation would also influence the relationship between FRC and SOC, given our hypothesis that deeper roots have a greater propensity for mineral interactions and thus contribute to persistent SOC pools. In our data we also observed generally deeper and more variable depth distributions of FRC and SOC in forests compared to grasslands (Figure S4). The variability in forest FRC-SOC relationships suggests that factors other than FRC contribute to forest SOC accrual to a greater extent than in grassland soils.

Third, more variable precipitation regimes in forests compared to grasslands could drive variability in the activity of decomposition enzymes and reactive metals. Thus, mineralogical limits to C storage, driven by variations in the amount and reactivity of clay minerals and reactive metals, may be more important controls on forest SOC than litter inputs^{31,32}. Since our predictive models do not include detailed proxies defining mineralogical limits to C storage, we may be missing some explanatory power and thus seeing high variability in forests. Furthermore, our grassland sites, on average, have half the mean annual precipitation of our forest sites. Thus, plant growth, litter inputs and decomposition could all be more moisture limited in grasslands than in forests, such that C inputs and not mineralogical limits are a more important factor for stabilizing SOC in grasslands.

...

Fifth, grassland plant roots typically have higher absorptive:transport root ratios and a greater proportion of absorptive roots with short lifespans, which could imply greater exudation and greater root litter contribution to SOC, respectively^{43,44}. Given the generally greater physiological activities of fine compared to coarse roots⁴³, a greater relative abundance of high-turnover, absorptive roots in grasslands compared to forests⁴⁵ might also result in greater activity rates of soil microbes. This greater microbial activity could also be higher in grasslands than in forests because grassland roots decompose faster than forest roots⁴⁶. We thus might also expect greater rates of microbial necromass production where fine root abundance is relatively greater, given fine root exudate activities and associated rhizosphere microbial growth and death⁴⁷. Evidence is accumulating that microbial necromass is a meaningful component of persistent SOC stocks⁴⁷⁻⁴⁹.

We might expect, then, that where absorptive root litter production is greater, necromass and thus SOC accrual might also be greater.

Quantitative comparisons between grassland and forest systems would further strengthen the results.

Response: We are unsure if this comment is for a specific section of the results/discussion. We already conduct quantitative comparisons between grasslands and forests, and these are presented in Figure 2 and 3, and their discussion, in the supplementary tables where models are presented with land cover type as a variable, and in Fig S4 where forest and grassland depth distributions are compared, including a new analysis that we added in the revision where we compare the depth at which 50% or 95% of the root biomass occur in forests and grasslands.

Additionally, the discussion on priming mechanisms in forests lacks clarity; more context and relevant citations should be provided to explain the observed variability. Expanding and deepening this section would significantly enhance the quality of the paper.

Response: We have edited this section extensively and have added three new paragraphs and several new references (please see the response above the last one).

Conclusions

The conclusion needs to be better integrated, summarized, and refined, particularly by emphasizing the most important findings of the study and their implications. Including recommendations based on the results would further increase the impact of this section.

Response: We have edited the conclusion and the limitation sections following the comments from this and other reviewers. Please see tracked changes file.

Materials and Methods

This section should be revised concisely and polished for better readability. For example, the phrase "low sample size" (L410) could be revised to "limited sample size" or "small sample size" for improved clarity and professionalism.

Response: fixed to "limited". Methods revised and streamlined in several places. Please see tracked changes file.

References

The reference format should be checked and revised for consistency. Ensure that all citations are accurate, uniformly formatted, and correctly aligned with the journal's guidelines.

Response: fixed several instances of inconsistencies.

In general, this manuscript addresses an important and timely topic in the field of ecosystem carbon dynamics. With the suggested revisions, the clarity, depth, and overall quality of the manuscript will be substantially improved, increasing its potential impact and suitability for publication.

Response: Thank you for your feedback and encouragement.

Dear Dr. Wang,

We thank you and the three reviewers for their thoughtful comments. We have now incorporated all remaining reviewer comments and conducted all checklist tasks. Below, please find the reviewer comments in unbolded text and our replies in bolded text.

**On behalf of all coauthors,
Avni Malhotra**

REVIEWERS' COMMENTS:

Reviewer #1 (Remarks to the Author):

This is a substantially revised manuscript. From my perspective, the authors have mostly addressed all of my previous comments. I have no further remarks at this time.

Reviewer #2 (Remarks to the Author):

The authors have satisfactorily addressed all of my concerns.

We thank reviewers 1 and 2 for their encouragement.

Reviewer #3 (Remarks to the Author):

This is a well-executed, conceptually rich, and increasingly polished manuscript. The revisions have successfully addressed most of the key issues from the previous round. I recommend acceptance after minor revision, mainly for clarity and consistency.

Thank you.

Specifically, the Results section emphasizes grasslands in all subtitles, while the Discussion includes only one forest-specific section without referring back to grasslands in the subtitle. I suggest improving the structural correspondence between Results and Discussion.

Great observation. We have now removed the single forest-focused subtitle in the discussion and have the following subtitles in the discussion:

- ***Higher belowground and absorptive root allocation in grasslands relative to forests***
- ***Variable depth distribution and chemistry of forest roots***
- ***The role of mineralogical limits and moisture availability differs between forests and grasslands***
- ***Mycorrhizal community complexity may be higher in forests than in grasslands***

In addition, please ensure consistency across the Abstract, Results, and Discussion, and refine the Conclusion to better highlight key insights and broader implications.

We edited the abstract, results, discussion and conclusion to ensure consistency. Notably, we added a paragraph at the beginning of the discussion to better tie the results and discussion, and to revisit hypotheses laid out in the introduction.

New paragraph: We found that grasslands drive the relationship between fine root and soil carbon stocks across a continental-scale observational gradient. Our hypotheses about the relationship between FRC and SOC were supported in grasslands but not in forests. Grassland FRC and SOC were strongly positively related, and SOC accrual was higher in high moisture and clay-rich grassland soils. Deep grassland soils had a particularly strong FRC-SOC relationship wherein net SOC accrual was more prevalent than priming. Conversely, forests showed high variability in FRC-SOC relationships. There are several possible explanations for why forest FRC-SOC relationships and priming may be highly variable compared to grasslands. In this section, we use existing literature to discuss possible mechanisms behind our observed differences in grasslands and forests.

Thank you to the authors for their thoughtful revisions.
Thank you for your time and valuable feedback.